# Rural Development Policy in Germany Regarding Coworking Spaces and Effects on Vitality and Versatility of Rural Towns

Marco Hölzel *[ID] and Walter Timo de Vries [ID]

School of Engineering and Design, Technical University of Munich (TUM), Arcisstr. 21, 80333 Munich, Germany; wt.de-vries@tum.de

* Correspondence: marco.hoelzel@tum.de; Tel.: +49-89-289-22565

**Abstract:** Remote rural areas have been declining in population for decades, partly permanently as people move away and partly temporarily, owing to commuting. This increasing paucity of inhabitants is causing these places to lose vitality and versatility; this, in turn, renders them less attractive overall. In terms of spatial development, policies devised for rural areas have long been concerned not only with agriculture, but also with holistic development. For some years now, ICT has work become increasingly location-independent. This is often seen as an opportunity for rural development. In addition to the general facilitation of remote working, i.e., working from home, coworking spaces make it possible to separate out our private and professional lives. The aim of this research is to find out to what extent public authorities position themselves on this topic and express their views on coworking spaces. Policies in this area have been promulgated by various federal ministries (Bundesebene) and federal states (Länderebene). Thus, we have collected relevant policies from the websites of federal ministries and three federal states (Bavaria, Schleswig-Holstein, Saxony-Anhalt), examined them for their keywords, and read and analyzed the documents that were found. Further, we have interviewed the founders and operators of particular coworking spaces. At the federal level, it is noteworthy that the ministry responsible for rural areas has published the greatest number of policies in which coworking spaces are mentioned. At the state level (Landesebene), the picture is more diverse, between the various state governments and the respective states. However, the contribution of coworking spaces to the vitality and versatility of rural towns is mentioned only rarely, and the importance of location is seldom pointed out. Comparing the results of this study with previous research in the literature, it can be concluded that public authorities should pay more attention to the opportunities and risks of coworking spaces in rural areas. Based on this, clearer objectives for coworking spaces in these areas can be formulated. When subsidies are disbursed, they should be accompanied by a mandatory evaluation to check what the subsidies have achieved and whether the subsidies have been used in a targeted manner. In addition, a larger database could be created for further research.

**Keywords:** rural development; public policies; vitality; coworking; location of work





## 1. Introduction

Changes in rural areas are part of a larger set of changes in socio-economic and spatial factors. Such a larger context also implies that research on rural development, as well as the design of rural development policies, must take such factors and the opportunities which may arise from them into account.

First of all, the practical effects of demographic changes [1] are much more significant for rural regions than for urban regions [2]. As it nevertheless is a political goal in Germany to obtain equal living conditions in both urban and rural areas [3] and to increase the attractiveness of rural regions to achieve this [4], a logical next step is that governmental bodies take a positive stance on the potential of coworking spaces to increase the attractiveness of rural regions, which is believed to have this effect. Regardless of whether this positive

stance is converted into any concrete financial support or is just a documented intention in a policy document, the mere fact of government agencies, especially ministries, speaking out on this coworking issue has a far-reaching effect on other agencies, authorities, and even commercial enterprises. It generates influence in the discourse on rural development. For this reason, it is important to detect how policies are executed. In other words, how government agencies approach the advocacy of rural coworking spaces in relation to overall rural development is the key objective of this study. Once this becomes more transparent, it can provide a practical key for the sustenance of rural coworking spaces in particular, as well as for rural development at large. We detail this main objective in detecting specifically how, where, when, and to which extent public authorities position themselves when expressing their views on coworking spaces.

Secondly, the theoretical discourse on rural equity and coworking spaces has also undergone several changes. The amount of literature on coworking spaces is rapidly growing, and an enormous pool of sources of varying quality is being created. In our preliminary literature review, we found several publications regarding public policies for coworking spaces in urban contexts [5–7], and a few regarding rural areas [8]. Some provided recommendations [9], and some were critical of government intervention [10]. What we did not find was a systematic analysis of public policies regarding coworking spaces and rural development. This paper attempts to fill this gap in relation to Germany and three federal states. In order to address any given grievances, different strategies can be pursued. Under specific economic and social conditions, administrative and legal frameworks, e.g., policies, have a significant effect on physical reality. While, in previous studies, we have tried to derive insights into empirical data, such as location information, from interviews and surveys, as well as from personal perceptions gained on site visits, in this study we endeavor to determine how policies and the insights from previous studies interact with each another.

Given the changes in both the practical and theoretical contexts, we rely on a pragmatist approach. This research approach aims at linking and aligning the findings from empirical research and the literature with overarching normative frameworks (i.e., local authorities, law, legislation). This is in order to make more precise observations on the one hand, and to align normative frameworks more precisely and expediently on the other. The process of cognition is an interactive hermeneutic process, as described by Gummesson [11], among others. Due to lack of data on measurable changes triggered by policies, we rely on the interpretation of texts discovered by keywords, the recommendations made in the policies, and predictions based on recommendations and assumed effects. Many considered policies are not developed as complex systems and thus lack an underlying theoretical foundation. An examination based on policy theories such as Hoogerwerf [12] analyses can offer little insight into the non-systematic and non-theory-based policies. The examined documents come from different institutions, often considering only individual aspects, and as such they are not complex policies for which political-theoretical analysis, as according to Harold Lasswell [13], is the aim of this study.

The subsequent sections of this paper are organized as follows. First, we take a look what can be found in the literature regarding current challenges for rural regions and towns (Sections 2.1–2.3). The second section of the literature focuses on knowledge work, which is (partially) location-independent, and on new workplaces (Section 2.3). Next, we shed light on the locations of new workplaces while taking research on the relationship between new workplaces and day-to-day amenities into account (Section 2.4). We also locate and situate our research in the context of the theory of knowledge (Section 2.5).

In the "Research Hypothesis" section, we formulate hypotheses and research questions. The "Materials and Methods" section describes the method which we used to obtain our data and the materials which served as the sources of our research. The next section describes the results we obtained from the sources. The "Discussion" section discusses our findings and the results of other research from the literature. The "Limitations" section

describes the limits of this research and what further research would entail. Finally, we formulate some recommendations for rural development policies.

## 2. Literature Review

### 2.1. Rural Decline and the Loss of Vitality and Versatility

Rural and remote regions in Germany, especially in the eastern part of the country, tend to suffer from out-migration [1,14]. Some people move to large cities [15] and so-called Schwarmstädte—swarm cities [16]—while others commute to urban settlements [17,18], or at least to office districts on the outskirts of cities. Both types of out-migrants are either less frequently or no longer present at all in rural areas and communities [19]. The absence of people in these places is accompanied by other losses. Shops and pubs make less money or have to close down [20]. Associations, including football clubs, voluntary fire brigades [21,22], and municipalities, receive less income or central funding [23]. Communication, exchanges, and a sense of community are all diminished by the reduced numbers of people in public spaces, in streets, and in squares, but also in the semi-public spaces of shops and pubs. It is mainly people of working age who are absent; the elderly have been left behind. Birth rates are often lower in remote regions [24], and, as schools with too few pupils are closed, these regions also become less attractive to couples who would like to have children [25]. All in all, this can be described as a pattern of demographic change. This shift is already taking place, and is even more pronounced in other countries, such as Spain, France, and Greece [26]. One of the main reasons for people moving away from empty hinterlands has been the greater number of jobs available in both capital and coastal regions.

With an absence of inhabitants, rural towns and communities lose their vitality and versatility. Both vitality and versatility are broad concepts that generally reflect socio-institutional and environmental characteristics, as well as construction and land use. In Germany, in a professional and academic context, vitality is defined as the multi-dimensional capacity of a predominantly educated community to sustain social, cultural, and economic activities; to retain a sense of independence and social cohesion; and to maintain long-term trust in the decision-making processes of relevant stakeholders. This definition relies on several sources, including the reflections of Schmied, Henkel and Miosga, Hafner [27,28]. Along similar lines, we can also derive a definition of versatility: versatility is based on a broad spectrum of cultural, economic, and social potential for change [29,30] and leads to a significant degree of autonomy and empowerment.

The effects of a decline in both vitality and versatility can be seen in out-migration. Those who move away leave behind them elderly people and other, less agile individuals. Population declines in rural towns and regions began with the industrialization of farming; this was due to the increased efficiency of agricultural production and the consequentially shrinking demand for an associated workforce [31,32]. This, in turn, exerted a kind of push–pull effect on the growing demand for workforce numbers in manufacturing and mining in cities and industrial regions, respectively, such as the Ruhrgebiet [33], a process which had already been underway, in any case [34].

People leave rural regions due to a lack of jobs [35] and educational opportunities [36], which cause losses of vitality and versatility in rural towns. Vitality and versatility are both regarded as indicators of attractive livability; accordingly, they have been investigated [37] and measured to evaluate the development trends of rural towns [38]. They are also currently the focus of ongoing research [37,39,40].

### 2.2. Former and Present Rural Development Strategies

Alongside the EU's direct agricultural policy, in recent decades, two strategies have been implemented to promote development and vitality in rural towns. One main strategy, as implemented by a group of policies, aims to attract industry and investment to rural regions by offering both public funding and cheap land for production plants, offices, logistical facilities, and public services [41]. Another principal strategy is designed to keep

people in rural towns or to attract them to rural areas; it consists of offering cheap building plots for detached housing. Both strategies result in a massive change in land use, of course [42,43]. Less densely built-up areas in peripheral locations also encourage greater use of cars and generate demand for street infrastructure, bypass roads, etc. At the same time, buildings in central urban locations increasingly fall into disuse or otherwise are used only partially [44,45]. With growing outskirts and declining central areas, towns are reshaped to acquire the shape of doughnuts, i.e., they have holes—or areas of emptiness—in their centers. This is the so-called Doughnut Effect [46].

### 2.3. Germany's Spatial Development Strategy

The Federal Spatial Planning Act (Raumordnungsgesetz) defines general aims of spatial development. The principles of spatial planning are described in its central section, § 2 para. 2 No. 1: "Balanced social, infrastructural, economic, ecological and cultural conditions are to be sought in the entire territory of the Federal Republic of Germany and in its sub-regions." [3]. In further sections of the Act, the factors of vitality and diversity are described as objectives of spatial planning, as well as of the economical, if not frugal, use of land.

### 2.4. Relocation of Work and Higher Education

In recent decades, the demand for skilled, highly educated workforces has only increased [47]. This process has been driven by academization [48] and is leading to both a knowledge-based economy [49] and a knowledge-based society [50]. In order to achieve the highest possible standard of living, young people are now orientating themselves to these requirements and planning their lives accordingly; thus, they strive for a high level of education and seek promising employment opportunities with successful companies that are easy to reach.

The increasing demand for higher education and superior job opportunities motivates young people to pursue academic degrees. In Germany, academic education is offered by 108 universities. These are located in major cities and some specific university cities [51], including universities of applied science and 106 universities with a specific focus on arts, religion, or pedagogy [52]. In order to obtain an academic degree, people have to move to these cities for the duration of their studies. This period of life is highly significant for many students, as it is then that they make new friends and form lasting relationships. Further, friendships, relationships, and job opportunities are the main factors influencing young people's choices of residence at the time of entering the job market [53].

This is why such companies tend to relocate to cities with universities or higher educational institutions, namely, in order to recruit graduates and high-potential employees [54]. Some companies which were formerly traditionally located in rural towns are now moving to major cities or opening branches in university cities [55].

### 2.5. Location-Independent Remote Working and Coworking Spaces

If a job is non-physical and can be performed solely based on knowledge and information or data, it can be performed from anywhere, provided that the white-collar worker has access to the required data and information. With the digitalization and the storage of knowledge in databases and digital libraries, only a log-in and an internet connection are needed [56,57] to perform knowledge-related work. With broader commercial use of computers and the impact of the oil crises in the 1970s, it became evident that it is cheaper and more fuel-efficient to transport bits and bytes through cables rather than to transport workers by road, or simply to "substitute transportation with communication"; a concept that had already been developed years before the oil crises [58]. Commuting is one of the things that people find tedious, as it makes them miserable [59,60] and causes mental and physical health problems [61].

The idea of remote working has been in existence for over 50 years, with occasional use by itinerant consultants, IT experts, and freelancers. Given the recent trends [62,63], current

research [64,65] is investigating the effects of people moving from urban to rural areas for residential and work purposes and how such persons contribute to the local communities and economies. IT experts, writers of all types, and freelancers were the first cohort to come together in coworking spaces [66], thereby "working alone together" [67]. Many of these knowledge workers experienced loneliness and isolation when working at home [68] became an available option.

Working from home can have other negative aspects as well, such as the additional burden of having to mix private life and work life in the same time and space. This dissolution of boundaries between professional and private life, known as "Entgrenzung," is fueled by the new flexibility of work brought about by advancements in information and communication technology (ICT) [59]. These challenges often affect women disproportionately, owing to traditional gender roles [69].

Research on coworking often investigates collaboration, co-creation, community, and the ways work is performed in such spaces [67,70–73]. Some studies focus on the physical space itself, including its interior design and health aspects [74–76], while others explore business models and trends [77,78], as well as real estate considerations [79,80]. The location of the coworking space is considered in some research, particularly in urban contexts [9,81,82], with some recent publications exploring coworking spaces in more rural environments and their potential in addressing demographic changes [83–87].

The COVID-19 pandemic highlighted the significant potential of remote work, particularly for knowledge-based work that does not require physical presence at a company site [88]. Although coworking spaces faced initial challenges due to pandemic-related restrictions, they were later recognized as a viable alternative for employees who no longer needed to commute to a company's remote office and who sought a professional, inspiring work environment separate from their private or family life [89].

In recent years, researchers [85], activists [90], think-tanks [84], and a growing number of politicians and administrators [91] have recognized the potential of coworking spaces as a way of increasing the attractiveness of rural and peripheral regions. However, there is limited research on the practical interactions of coworking spaces with their (immediate) surroundings and how they might contribute to the vitality and versatility of adjacent areas and/or towns and villages.

*2.6. Vitality by POIs (Points of Interest)*

The various research projects on vitality and versatility have not provided any final or definitive assessment; despite this, it is recognized that vitality and versatility relate to both people and activities [37,40,92]. In the case of biological organisms, vitality is defined by the presence of a metabolism, which involves the exchange of substances and, in part, information. This concept also applies to cities, villages, and settlements, i.e., places where perceptible exchanges and encounters occur, and where people come together, are often referred to as vibrant or vital. For people to be able to interact with each other, they need to be physically present near each other, rather than simply commuting to a town, city, or the outskirts of a village, or even just relocating.

Points of interest (POIs) related to daily life, such as grocery stores, restaurants, bakeries, and child-care facilities, generate physical activities [93] similar to multi-purpose trips [94], which aligns with the concept of the "15-Minute City" [95]. The more gatherings and interactions are possible, the denser the network of relationships and exchanges becomes, with this process resembling the workings of a vital organism. In addition to physical activities, these POIs offer opportunities for both affective and intellectual engagement with others. When work is performed from a coworking space in the vicinity of a place of residence, the relevant commute can then be accomplished on foot or bicycle, thereby saving fuel, reducing $CO_2$ emissions, keeping people healthier, and the increasing capacity of the town, as well as promoting physical and emotional interaction in the vicinity of individuals' places of residence [93,96].

## 3. Research Hypotheses

Rural development policies have traditionally focused on the agricultural sector, especially small farms, as these are predominant in rural regions when viewed overall [97].

The reality of rural regions is, however, that the influence of the agricultural sector—and the role of agricultural smallholdings in particular—has gradually decreased in rural areas. Instead, rural areas have either become more diverse in terms of function and operation (thriving rural villages) and/or have increasingly developed farming systems owned and operated by companies with multiple business sites. Additionally, rural regions have developed new requirements, which are no longer exclusively agricultural. The consequence of this is that such policies are no longer appropriate for developing or sustaining rural regions. Hence, there is a need for a critical discussion on how and where such policies are appropriate, in view of the new needs of rural regions.

In Germany, rural development policies after the Second World War must be viewed in the context of the EU. Since the founding of the European Economic Community (EEC), the agricultural sector has been a major issue. The CAP (Common Agricultural Policy) has dominated EU policies since the 1960s [98]. In the wake of general market liberalization in the 1980s, the price stabilization policy was abandoned, and since the 1990s, the second pillar of the EU agricultural policy was developed and expanded. Starting in 1991, the LEADER (French acronym: Liaison entre actions de développement de l'économie rurale) program was initiated. The operation of this program was considered successful, and it was decided that it should be continued [99].

If rural development policies are intended to target the vitality of rural towns by supporting coworking spaces, they need to consider human interactions and their likelihood of occurring; this is clearly increased by the potential venues for such encounters. These would include points of interest (POI), amenities, services, and other opportunities for people to meet each other. We took the previous sentences as a starting point for our hypotheses and posed the following sub-questions:

1. How do policies or programs address coworking spaces? We anticipate that there are various policies and programs which are partly overlapping and partly contradicting.
2. Can these policies and programs be distinguished in terms of spatial types? We hypothesized that the spatial nature is not always given, and therefore spatial inequity remains insufficiently addressed in practical policy implementation.
3. How many policies take the contribution of coworking spaces to vitality into account? We assumed that whilst coworking spaces are an item in some polices, the actual impact remains insufficient to make the policies sufficiently effective.
4. Do these policies specify preferential locations? We assumed that certain preferences would be present, but perhaps not transparent enough.

## 4. Materials and Methods

The freedom to choose one's place of residence—Freizügigkeit—is a main aspect of freedom in modern societies. Because of such free movement of persons, there is no possibility for governments to allocate sections of the population to specific places of residence. The only way for them to intervene in this area is by creating a gentle push–pull effect, as some policies do. Thus, a growing number of policies address the attractiveness of rural regions [100–107], and, in recent years, an increasing number of policies and strategies have been developed regarding coworking.

To answer these research questions, we relied on policies found on the websites of policy-making institutions to develop a systematic structure for our study.

We considered any written statement from these policy-making institutions to be a policy. Policies relevant to rural development were mainly found at three administrative levels: first, the EU (European Union) level; second, the national or federal level; and third, the state, sub-national, or regional level.

The policies discovered by a keyword search on the websites of the institutions identified below were read and analyzed according to the following parameters: (i) whether

they address rural or urban locations, (ii) whether they address the fact that such uses can contribute to the vitality of places, and, in particular, public spaces; and (iii) whether these sources make recommendations as to which location is preferable.

## 5. Results

### 5.1. European and National Level

Policies from the European Parliament or the European Commission usually need to be implemented at the national level in order to be enacted and effective [108]. Depending on the national structure involved, sub-national structures may need to be authorized to implement these policies. It is, therefore, the national level that is most relevant, as it is here that EU regulations are transposed into national law and the underlying state structures can be mandated and rendered accountable.

### 5.2. National Level

In this phase, we began our research at the national level. We searched the websites of all 15 federal ministries for policies, programs, and other written documents using the search term "coworking" for the systematic research. The occurrence of the search term in documents was used to narrow down which documents to examine in more detail.

The next step was to read the documents and texts and to analyze them. As a third step, we looked for three categories: (i) Are urban, rural, or both settings addressed? (ii) Is there a mention of vitality or similar being addressed? (iii) Are there some recommendations made regarding the locations of coworking spaces?

In Table 1, we show, in the first column, the number of hits for the search term; in the second column, the number of hits for the type of location or settlement structure; in the third column, the number of hits for vitality or similar; and in the fourth column, the number of hits for location or similar.

**Table 1.** The federal ministries, showing the search results for the term "coworking" on the ministries' websites. This table also shows whether the reference was to rural, urban, or both locations; whether vitality or similar was mentioned; and whether a location recommendation was made.

| Federal Ministry | Search on Website with Keyword | (i) Regarding: Rural (1), Urban (2), Unspecified (3) | (ii) Regarding Vitality or Similar | (iii) Recommendations |
|---|---|---|---|---|
| Federal Ministry of Economics and Climate Protection (BMWK) | 3 | $1 \times 1, 1 \times 2, 2 \times 3$ | 1 | 0 |
| Federal Ministry of Finance (BMF) | 0 | | | |
| Federal Ministry of the Interior and Home Affairs (BMI) | 5 | $3 \times 1, 3 \times 2$ | 4 | 5 |
| Federal Foreign Office (AA) | 0 | | | |
| Federal Ministry of Justice (BMJ) | 0 | | | |
| Federal Ministry of Labour and Social Affairs (BMAS) | 2 | $1 \times 1, 1 \times 2$ | 1 | 1 |
| Federal Ministry of Defense (BMVG) | 0 | | | |
| Federal Ministry of Food and Agriculture (BMEL) | 18 | $17 \times 1, 1 \times 2, 1 \times 3$ | 8 | 5 |
| Federal Ministry for Family Affairs, Senior Citizens, Women and Youth (BMFSFJ) | 7 | $1 \times 1, 6 \times 3$ | 0 | 0 |
| Federal Ministry of Health (BMG) | 0 | | | |
| Federal Ministry of Digital Affairs and Transport (BMDV) | 0 | | | |

**Table 1.** *Cont.*

| Federal Ministry | Search on Website with Keyword | (i) Regarding: Rural (1), Urban (2), Unspecified (3) | (ii) Regarding Vitality or Similar | (iii) Recommendations |
|---|---|---|---|---|
| Federal Ministry for the Environment, Nature Conservation, Nuclear Safety and Consumer Protection (BMUV) | 1 | $1 \times 1$ | 0 | 0 |
| Federal Ministry of Education and Research (BMBF) | 2 | $1 \times 1, 1 \times 3$ | 0 | 0 |
| Federal Ministry for Economic Cooperation and Development (BMZ) | 1 | 3 | 0 | 0 |
| Federal Ministry of Housing, Urban Development, and Construction (BMWSB) | 5 | $2 \times 1, 4 \times 2, 1 \times 3$ | 1 | 2 |
| Total | 44 | | 15 | 13 |

*5.3. Sub-National Level—Federal States (Bundesländer)*

The German Bundesländer are responsible for implementing EU policies and programs, for which they have been authorized by the Federal Parliament. Of the 16 German Bundesländer, three are city-states, which are less relevant to rural development policy. Of the remaining thirteen Bundesländer, we selected three: one in the north (Schleswig-Holstein), one in the south (Bavaria), and one in the center of Germany, formerly part of the GDR (Saxony-Anhalt).

We used the same system at the federal level, searching the websites of the three state governments for policies, programs, or other written documents and texts using the search term "coworking".

It should be mentioned that the structures of the three state governments' websites differ greatly. Similarly to the federal level, all ministries in Bavaria have their own websites with individual search functions. In Schleswig-Holstein, there is only one state government website with all ministries represented, while in Saxony-Anhalt, the ministries have their own websites, although the search function is provided by a central server.

The occurrence of the search term in specific documents was used to narrow down any documents that need to be examined in more detail. The next step was to read the documents and texts to see what they were about and to analyze the overall context. In a third step, we looked for three categories: (i) Are urban, rural, or both settings addressed? (ii) Is there a mention of vitality or a similar term? (iii) Are some recommendations made regarding the location of coworking spaces?

In Table 2, we show a list of Bavarian ministries. In the first column are the number of hits for the search term; in the second column, how often the type of location or settlement structure was addressed; in the third column, how often vitality or similar was addressed; and in the fourth column, how often the documents mentioned or gave some hints regarding the location.

Table 3 shows the number of web pages/documents of the government of Schleswig-Holstein in which the search term was mentioned. The second column shows the number of hits for the search term in all of the documents which were found. The third column shows how often the type of location or settlement structure is mentioned. The fourth column shows how often vitality or similar was mentioned, and the fifth column shows how often the documents mentioned or gave some indication of the location.

**Table 2.** The Bavarian State Ministries are listed, showing the search results for the term "coworking". The table also shows whether the reference was to rural, urban, or both locations; whether vitality or similar was mentioned; and whether a location recommendation was made.

| Ministry | Search on Website with Keyword | (i) Regarding: Rural (1), Urban (2), Non-Specific (3) | (ii) Regarding Vitality or Similar | (iii) Recommendations Regarding Location |
|---|---|---|---|---|
| Bavarian State Chancellery | 14 | $2 \times 1, 2 \times 2, 9 \times 3$ | 0 | 0 |
| Bavarian State Ministry of the Interior for Sport and Integration | 0 | | | |
| Bavarian State Ministry of Housing, Construction and Transport | 5 | $1 \times 1, 2 \times 2, 2 \times 3$ | 0 | 0 |
| Bavarian State Ministry of Justice | 0 | | | |
| Bavarian State Ministry of Education and Cultural Affairs | 0 | | | |
| Bavarian State Ministry of Sciences and the Arts | 0 | | | |
| Bavarian State Ministry of Finance and Home Affairs | 2 | $0 \times 1, 0 \times 2, 2 \times 3$ | 0 | 0 |
| Bavarian State Ministry of Economic Affairs, Regional Development and Energy | 9 | $3 \times 1, 0 \times 2, 6 \times 3$ | 1 | 0 |
| Bavarian State Ministry for the Environment and Consumer Protection | 1 | $0 \times 1, 0 \times 2, 1 \times 3$ | 1 | 1 |
| Bavarian State Ministry for Food, Agriculture and Forestry | 0 | | | |
| Bavarian State Ministry of Family, Labour and Social Affairs | 0 | | | |
| Bavarian State Ministry of Health and Care | 0 | | | |
| Bavarian State Ministry of Digital Affairs | 2 | $0 \times 1, 0 \times 2, 2 \times 3$ | 0 | 0 |
| Total | 33 | $6 \times 1, 4 \times 2, 22 \times 3$ | 2 | 0 |

**Table 3.** This shows the search results for the term "coworking" on the website of the government of Schleswig-Holstein. In addition, the table shows whether the reference was to rural, urban, or both locations; whether vitality or similar was mentioned; and whether a location recommendation was made.

| Federal State Government of Schleswig-Holstein | Search on Website with Keyword | (i) Regarding: Rural (1), Urban (2), Unspecified (3) | (ii) Regarding Vitality or Similar | (iii) Recommendations Regarding Location |
|---|---|---|---|---|
| 38 | 111 | $17 \times 1, 0 \times 2, 21 \times 3$ | 11 | 9 |

In the case of Schleswig-Holstein, we spoke to the responsible persons at the Ministry of Agriculture, Rural Areas, Europe and Consumer Protection (Ministerium für Landwirtschaft, ländliche Räume, Europa und Verbraucherschutz) about the objectives, funding criteria, responses, and evaluation of the funding program for coworking spaces in Schleswig-Holstein.

Table 4 shows the number of websites of the government of Saxony-Anhalt. The first column shows the number of websites on which the search term could be found. The second column shows the number of hits for the search term on all websites. The third column shows how often the type of location or settlement structure was addressed. The fourth column shows how often vitality or something similar was addressed in the documents, while the fifth column shows how often the documents mentioned or gave hints regarding location.

**Table 4.** The search results for the term "coworking" on the website of the government of Saxony-Anhalt. The table also shows whether the reference was to rural, urban, or both locations; whether vitality or similar was mentioned; and whether a location recommendation was made.

| Federal State Government of Saxony-Anhalt | Search on Website with Keyword | (i) Regarding: Rural (1), Urban (2), Non-Specific (3) | (ii) Regarding Vitality or Similar | (iii) Recommendations Regarding Location |
|---|---|---|---|---|
| 41 | 143 | $21 \times 1, 4 \times 2, 16 \times 3$ | 5 | 3 |

*5.4. National/Federal Level (Bund)*

The keyword search on the websites of the different national and federal ministries (*Bundesländer*) at the administrative level provided a rough overview of how present and relevant coworking and coworking spaces are in the policies of the national and federal governments.

At the national/federal level, we found 44 documents in which coworking was mentioned. Depending on the responsibilities of the ministries, few (1–3), some (5–8), many (18), or even no mentions were found. It was to be expected that no statements on coworking were found in some federal ministries; e.g., the ministries of finance, defense, foreign affairs, and justice. The Ministry of Labour only mentioned coworking in two documents, and the same applies to the Ministry of Education and Research. The Federal Ministry of Economics and Climate Protection mentioned coworking in three documents. The Federal Ministry of the Interior and Home Affairs and the Federal Ministry of Housing, Urban Development, and Construction mentioned coworking in five documents each. The Federal Ministry for Family Affairs, Senior Citizens, Women, and Youth mentioned coworking seven times, mainly in the context of childcare. The Federal Ministry of Food and Agriculture (BMEL) mentioned coworking 18 times, with a clear focus on rural settlements.

Vitality and similar concepts are shown only once each on documents from the Federal Ministry of Economics and Climate Protection; the Ministry of Labour and Social Affairs; and the Federal Ministry of Housing, Urban Development, and Construction. The Federal Ministry of the Interior and Home Affairs mentioned vitality or similar concepts four times, and the Federal Ministry of Food and Agriculture eight times. Considering the large number of documents from the Federal Ministry of Food and Agriculture, it seems reasonable for vitality and similar concepts to have been mentioned more often. It also seems reasonable that the demand for vitality in regions dominated by agriculture and for which the Ministry of Food and Agriculture is therefore responsible manifests both a lower level of vitality and a higher demand for the same. Both arguments may be based on the same fact: there is a distinct lack of vitality in rural towns and regions, which is why the ministry responsible has published a significant number of documents on coworking, specifically reflecting on vitality within them.

*5.5. Regional/State Level (Bundesländer)*

The three states which we investigated (Bavaria, Saxony-Anhalt, and Schleswig-Holstein) presented very different pictures. For Bavaria, we were able to identify the ministry responsible due to the existence of separate websites, whereas for Saxony-Anhalt and Schleswig-Holstein, this was not always possible.

On the website of the Bavarian State Chancellery, as the concluding unit, we managed to find the search term 14 times; on the website of the Bavarian State Ministry for Economic Affairs, Regional Development, and Energy, nine times; on the website of the Bavarian State Ministry of Housing, Construction, and Transport, five times; on the website of the Bavarian State Ministry of Finance and Home Affairs, twice; and on the website of the Bavarian State Ministry for Digital Affairs, twice as well. On the website of the Bavarian State Ministry for the Environment and Consumer Protection, the search term could be found only once. On the websites of the other seven ministries, the search term could not be found at all; this includes the Bavarian State Ministry of Food, Agriculture, and Forestry, which is responsible for rural regions. Vitality and similar concepts were mentioned only twice in total.

A search conducted on the Schleswig-Holstein government website yielded 38 documents, in which the search term was found 111 times. Vitality or a similar term was mentioned eleven times, while location recommendations were found nine times. Regarding Schleswig-Holstein, it is worth noting that several of the search results dealt with a funding program for coworking spaces in rural areas.

A search on the website of the government of Saxony-Anhalt identified 41 documents, in which the search term was found 143 times. Vitality or a similar term was mentioned five times, and location recommendations were found three times. The situation for Saxony-Anhalt was the same as that for Schleswig-Holstein; we found several documents dealing with a funding program for coworking spaces in rural areas.

Comparing the three federal states of Bavaria, Saxony-Anhalt, and Schleswig-Holstein, it can be seen that, at the time of the survey, different levels of attention were given to the topic of coworking in each of these states. In addition to the very different frequencies of documents dealing with the topic, the respective intensity of treatment in these documents also differed, while the factors they considered were also distributed differently: although rural areas were mentioned relatively often, the contribution of CWS to vitality was not mentioned as often as it should have been. Further, recommendations for site selection were also rare.

### 5.6. The Perspective of Coworking Space Owners and Operators

To obtain the perspective from the other side, we talked to three owners and operators of coworking spaces in Bavaria, Schleswig-Holstein and Saxony-Anhalt, taking into account the following: versatility; vitality; government support, e.g., business/startup/management consulting, funding, etc.; what they receive from the government; and what they would like to request from the state and federal governments.

Owners and operators of coworking spaces in Bavaria, Schleswig-Holstein, and Saxony-Anhalt responded more or less with one voice: they mentioned that they believed their coworking spaces contribute to the diversity and vitality of their areas by (1) bringing unused buildings back into use; (2) bringing people to the venue, which ensures the presence of people in adjacent streets by coming and going, taking breaks, having lunch, etc.; and (3) contributing spending capacity to the local area. Our interview partners did not see any general requirement for governmental support, such as the provision of advice, etc., as opposed to actual funding, which, in fact, only a minority of them have received so far. Support or advice on how to apply for funding is also seen as helpful, however.

## 6. Discussion

Rural development policies attempt to keep rural areas and towns attractive or, at least, to prevent them from losing their attractiveness for people and businesses [41]. A measure designed to attract people and businesses is to zone land for development. Through this process, former agricultural land is often converted into building land and then ceases to be available for agricultural use. This does not seem to be a sustainable pathway for future

development, especially if land consumption continues to be well above the target of 30 ha (federal territory) [43].

The idea of remote working via the transportation of data, rather than people, has existed for over 50 years [58]. With the development of computer technology and ICT in recent decades, this concept has become increasingly relevant [56]. The interpersonal contact restrictions imposed during the COVID-19 pandemic have shown just how much knowledge-related work can be performed remotely [88].

In addition to the policies that address the attractiveness of rural areas [100–107], a growing number of policies also address coworking in general, and in rural areas in particular. Policies regarding coworking in rural regions at the federal level have mainly been initiated by the Federal Ministry of Food and Agriculture (BMEL). The national policies which we investigated are listed in Table 1. At the level of the federal states, a different picture emerges: while in Bavaria, the search term could not be found on the website of the ministry responsible for rural development, i.e., the Bavarian State Ministry for Food, Agriculture, and Forestry. In Schleswig-Holstein and Saxony-Anhalt, in contrast, several documents offering multiple hits for the search term could be found. It seems that Schleswig-Holstein and Saxony-Anhalt are much more aware of the issue than Bavaria. This is reflected, on the one hand, by the greater number of relevant documents, which also deal with the conditions much more intensively, but also by the fact that Schleswig-Holstein and Saxony-Anhalt offer funding programs expressly for coworking spaces.

The question of where a coworking space should be located, or of which location offers more opportunities, still seems to be of great relevance with regard to the relationship between vitality, social interaction, and the possibility of their enhancement through personal interactions [9,82,93]. Research on coworking has mainly focused on coworking spaces in urban settlements [9], although in recent years, researchers have increasingly considered the spatial distribution of coworking spaces [81,82]. Therefore, we believe that it is important to consider these factors in policy making.

As previous research has shown, the locations of coworking spaces seems to be relevant to the impact of these spaces on the vitality of rural towns [82,93].

Only 15 out of 44 policies at the federal level mentioned the effect of coworking spaces on vitality or similar (34%). Of the 18 policies launched by the Federal Ministry for Food and Agriculture, however, 8 mentioned the effect of coworking spaces on vitality (44%). The influence of coworking spaces on vitality in their vicinity could attract more awareness to this issue in public policies.

Out of the 44 policies at the federal level, 13 give recommendations regarding the location of coworking spaces (30%). Of the 18 policies of the Federal Ministry for Food and Agriculture, five give recommendations regarding the location of coworking spaces (28%). Just a few policies mention the locations of coworking spaces. This aspect has the potential to gain more attention due to the high level of interaction between coworking spaces and their neighborhoods.

The picture is more varied at the level of the Bundesländer. While, in Bavaria, there are relatively few policies/documents that deal with coworking spaces, even marginally, the issue seems to be much more evident in Schleswig-Holstein and Saxony-Anhalt. There are even specific subsidies for coworking spaces. It is particularly surprising that we found not even one document in which the Bavarian ministry of agriculture, which is responsible for the development of rural areas, addresses this issue. The Free State of Bavaria likes to see itself as an innovative and high-performance location, but in the context of coworking spaces, however other federal states, are closer to the pulse of time, current developments, and innovation.

Even if coworking space operators and owners do not have explicit demands other than for financial support, policy makers should think carefully about what they are trying to achieve, how they want to manage whatever processes they may instigate, and, at the very least, what recommendations they would like to pass on to those operators or other organizations that would like to run their own campaigns in these areas.

Coworking space operators are less interested in advice from authorities, which would probably be perceived as a form of paternalism. However, operators are usually interested in financial support or at least in advice regarding how and where they can best apply for funding.

For potential funding providers, however, it is, of course, important that the granted funds are used precisely for the purpose for which they are intended. There is not always an overlap here, because the goals only partially overlap. The objectives of the policies are often formulated in a very abstract way and can be interpreted in different manners. This provides the advantage that specific constellations can be addressed in individual cases. However, a clear description of the aim of developing coworking spaces in rural areas could contribute to a clearer picture and fewer misinterpretations on the side of the operators and founders of coworking spaces. A more concrete description of the goal of sustainable rural development could emphasize aspects such as shorter commutes, community building both in the coworking spaces and within their vicinities, retention of purchasing power, and fostering vitality, as described in more detail below.

With regard to the general policy goals of revitalizing rural towns and reducing land consumption [43], it seems necessary to choose a location for a coworking space that is not in an industrial or commercial area on the outskirts of town or on a bypass road, which, in practice, is sometimes the case [85,93]. It seems more advantageous to choose a location in the center of a town, and, thus, to make use of the existing building stock. This would provide easy access to other amenities in the area, such as supermarkets, childcare, shops, restaurants, public transport, etc. [93], and would help individuals to combine trips to their places of work with other purposes. This would allow for multi-purpose trips [94], with the overall result being consistent with the concept of the "15-Minute City" [95].

### 6.1. Conclusions

Referring to the research (sub-)questions, and based on the findings, we can derive the following answers:

We found 44 policies at the federal level, 33 within the Bavarian State, 111 in the state of Schleswig-Holstein, and 143 in the state of Saxony-Anhalt. It is, therefore, not a surprise that these respective policies are not always in sync, and therefore are sometimes contradicting in their content and their aims. This confirms our first hypothesis.

1. Regarding the distinction between spatial types, we have to admit that here, no clear picture could be obtained. In Saxony-Anhalt, a majority of policies considered rural settings. In Schleswig-Holstein, a majority of policies neglected to consider the spatial setting, which was also the case in Bavaria. At the federal level, there was majority of rural settings mentioned in the policies due to the strong orientation of the Federal Ministry of Food and Agriculture towards coworking spaces in rural areas.
2. As the distinction of policies and programs in terms of spatial types is concerned, the results demonstrate that the spatial nature is not always a given in any of these policies. In other words, this confirms our hypothesis that the policies are insufficiently taking the spatial variation of certain aspects into account, which, in turn, prevents them from sufficiently contributing to spatial equity.
3. Regarding whether policies take the contributions of coworking spaces to vitality into account, we observed that vitality was indeed mentioned on the federal level by 15 policies, but in Bavaria only by 2, in Schleswig-Holstein by 11, and in Saxony-Anhalt by 5. Vitality is, in other words, not coherently or systematically addressed, and is therefore not (yet) a sufficiently effective key element in rural development policies.

Finally, as far as preferential locations are concerned, we found that preferential locations were mentioned at the federal level by 13 policies, in Bavaria by 0, in Schleswig-Holstein by 9, and in Saxony-Anhalt by 3. This confirms our hypothesis, namely, that certain preferences would be present, but perhaps would not be transparent enough.

Given these findings, we can conclude that the political bodies are indeed aware of the relevance of coworking spaces for rural development, given the documented artefacts and

given the positive framing of the relevance and significance of these. Nevertheless, since just a few detailed documents were found, we obviously still see gaps in the ministries' knowledge and we would argue that there still exists at least a lack of clarity regarding what is to be welcomed or supported.

Some states (Schleswig-Holstein, Saxony-Anhalt) are engaged in intensive examinations of this issue, partly with their own programs, with which they gain experience and an ability to more safely handle of the phenomenon.

Despite these conclusions, we are aware of some limitations of the execution of this research. Unfortunately, our online keyword searches provided no guarantee that all documents issued by the relevant local authorities could be found. Hence, there may be other policies on coworking spaces at the regional or county level which we were unable to discover. We did not include these, quite simply because the sheer number of them is gigantic. Furthermore, there may be other policies or funding schemes that could potentially be applied to coworking spaces, but these are not mentioned specifically; these were not included due to possible variations in interpretation.

Although we visited many of the coworking spaces, we were unable to make any comparisons between the levels of vibrancy or vitality of the spaces before they were used as coworking spaces or for their current uses. This is because it was only possible to identify the relevant locations in our online searches after the time at which they were reused as coworking spaces.

Further research should empirically analyze the effect of coworking spaces on vitality and versatility. Funding for rural development should also be evaluated as a potential source of empirical data that could potentially overcome the inherent limitations of research, or, at least, expand its fact base.

*6.2. Recommendations*

The following concrete recommendations can be derived from the results of the discussion, taking into account the correlations between place of work and vitality/versatility:

A:   Considering the basic goals of spatial development—equal living conditions in all subspaces and an economical use of land—coworking spaces should be located in rural regions in primarily central locations of villages and cities. This is in order to increase vitality and versatility in places where the vitality is perceptible.

B:   These preferential locations should be promoted by appropriate policies of governments, ministries, and other institutions. These should be designed to achieve the intended effects and to avoid unwanted effects, such as increased land use, traffic, etc. The role model function of a governmental or ministerial policy/recommendation must also be considered here. A precise description of what institutions and authorities are aiming for in the development of rural villages and towns facilitates the reconciliation or balancing of societal and individual interests.

C:   Subsidies should be linked to compliance with the above recommendations in order to focus the use of public funds and avoid misdirected investments.

In order to make more reliable recommendations, further research should be conducted on coworking spaces and the interactions of their users with their environs. This should consider both economic and social factors. Furthermore, funding for coworking spaces should finance a mandatory evaluation of the above interactions. At the very least, this evaluation should recommend modest, low-key policies.

The next step in following up on this research would be to start a dialogue between policy makers and academics, which would include examining the opportunities for the further development of coworking. Such a dialogue in states with a high demand, such as Schleswig-Holstein, might be a good first step.

Depopulated regions in France, Spain, and Greece are facing more or less the same demographic developments as those in Germany. Thus, findings from Germany can be applied to those regions. One of the main reasons why people have moved away from depopulated hinterlands is the greater number of jobs available in the respective capital

and coastal regions. Coworking spaces could relocate these places of work back to rural regions.

Based on the results of this research, it should be investigated which policies have actually been implemented and, in particular, exactly how the funding programs of Schleswig-Holstein and Saxony-Anhalt have been used.

Furthermore, it should be investigated which further sources of funding for coworking spaces have been utilized. In both cases, it would be desirable to investigate what requirements or pre-conditions are attached to this funding. It would also be interesting to ascertain what interactions coworking spaces and their users have with their respective neighborhoods; how the use of coworking spaces changes people's daily routines; and just how this process contributes to the vitality of local communities and their public spaces.

**Author Contributions:** Conceptualization, M.H.; methodology, M.H.; investigation, M.H.; data curation, M.H.; writing—original draft preparation, M.H.; writing—review and editing, M.H. and W.T.d.V.; supervision, W.T.d.V. All authors have read and agreed to the published version of the manuscript.

**Funding:** This research received no external funding.

**Institutional Review Board Statement:** Not applicable.

**Informed Consent Statement:** Not applicable.

**Data Availability Statement:** Additional data which were analyzed in this article can be retrieved upon request.

**Conflicts of Interest:** The authors declare no conflict of interest.

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
