# Peer review of "Rural Development Policy in Germany Regarding Coworking Spaces and Effects on Vitality and Versatility of Rural Towns"

_urbansci, doi:10.3390/urbansci7030086_

Round 1

Reviewer 1 Report

Please supplement the clear purpose of the study and further revise the practical implications through the results of this study.

Author Response

We try to describe the purpose of the study more clearly and added practical implications.

Reviewer 2 Report

The paper needs more depth. 

1.      The abstract section must include research objectives and conclusions in the content. It is recommended to add these aspects for clarity.

2.      To improve focus and clarity for readers, it's recommended to reduce the keywords from 10 to no more than 5, to make the main points more apparent and easier to understand.

3.      In the introduction, it is important to state the research objectives mentioned in the abstract. This section should also provide an overview of the current progress in the international research field while identifying any gaps or limitations that must be addressed. Please ensure that the literature is thoroughly reviewed and integrated.

4.      The article’s research significance and the problems it aims to address are not clearly stated. It would be helpful if the author could provide additional information in this section of the content.

5.      Please add an introduction to the research topic in the article. Additionally, is the research methodology used appropriately?

6.      The conclusion section mainly consists of a list of materials. This part should be concise and clear, providing creative, instructive, and empirical descriptions of the results.

Minor editing of English language required.

Author Response

To 1.

We added the research objectives and some conclusions from the main text to clarify that already in the abstract.

To 2.

The topic of this article is related to many different research fields, therefore more keywords could attract more researches from other research fields to this article. After all we reduced the number of keyworks to meet this demand.

To 3.

We added the research aim in the introduction. The current status of the international research on coworking spaces is described as far as known. If some important source is missing please share a link with us.

To 4.

We hopefully described more clearly what the challenges are and why this study is relevant.

To 5.

We have added an explanation of the research topic. We address the appropriateness of the methods and their correct application in the limitations.

To 6.

We separate the more descriptive results from our study in the section ‘Results’, we draw conclusions by discussing the study results with the insights from the literature in the section ‘Discussion’ where we provide conclusions by linking and interpreting empirical insights and literature. In the recommendations section, we derive suggestions from the results of the studies and their discussion in the discussion section and formulate future research needs.

Reviewer 3 Report

The authors have presented an insightful examination of how coworking spaces can potentially revitalize remote rural areas, a topic of considerable relevance and interest. The manuscript delves into how the introduction and enhancement of ICT, as well as the facilitation of remote working, could potentially counteract the population decline and loss of vitality observed in rural regions.

The authors' methodology appears to be sound and comprehensive, with the data collected from various federal and state ministries, and their policies towards rural development and coworking spaces. The additional inclusion of interviews with founders and operators of coworking spaces is an excellent qualitative supplement to the policy analysis. This provides a well-rounded perspective on the issue at hand.

The study's findings highlight interesting discrepancies at the federal and state levels, adding further nuance to the discussion. This discrepancy and its implications could be explored in greater detail in the main body of the manuscript.

Furthermore, the authors appropriately acknowledge the potential limitations of the existing policies, particularly the lack of emphasis on the contribution of coworking spaces to the vitality of rural areas and the importance of location. This critique sets a stage for a deeper discussion and potential suggestions for policy modifications.

Decision:

In view of the significant contribution this manuscript makes towards understanding the role of coworking spaces in rural revitalization, I recommend acceptance of the manuscript. The authors have skillfully demonstrated a complex issue in a detailed and engaging manner. The paper is likely to stimulate further research and policy debate on this subject. I look forward to seeing this paper published and shared with the wider academic community.

Author Response

Thank you very much for this detailed review, which enriches us with good suggestions and motivates us to further expand our research.

We have broadened the discussion to include the perspective of where the problems of the current situation lie and what should be done about them.

Reviewer 4 Report

- The article is welcome to the Urban Science Journal.

- Please check if No. 12 - Line 964 in References List is mentioned in the article's text

Author Response

Thank you for welcoming us to the Urban Science Journal J

There was a malfunction in the citation program, which we were able to fix. Now the citations and sources are correct again.

Reviewer 5 Report

Dear authors,

Very interesting study. Only some aspects must be reviewed:

-it is a little bit too long, the reader is bored

-in the introduction part you must reduce the numbering: 1.1….12.1.3…it is too long the introduction part

-the result part is almost inexistent ….is shorten then introduction and methodology and it must be the most important part

In my opinion, the work does not fulfill the conditions of a work with visibility, at least in this form, therefore I am not in favor of publishing the work.

Author Response

Thank you for considering our research to be very interesting. We are very pleased.

We propose to reduce the overall length of the manuscript by omitting the appendices.

Numbering helps to structure the complex interrelationships and to orientate the reader in the text, so we would like to stick to it.

We have expanded the results section of the manuscript. This is of course the most important part.

It is a pity that you do not want to publish the manuscript. We only received your comments after the comments of the other reviewers when we resubmitted or manuscript. We believe that the current revision, addresses all comments of the other reviewers as well as the main part of your comments.

Round 2

Reviewer 2 Report

The article has been better modified.

Author Response

Thank you for your kind feedback that we have improved the manuscript.

Reviewer 5 Report

Dear authors,

Very interesting study. Only some aspects must be reviewed:

-in the introduction part you must reduce the numbering: 1.1….12.1.3…it is too long the introduction part

-the result part is almost inexistent ….is shorten then introduction and methodology and it must be the most important part

In my opinion, the work does not fulfill the conditions of a work with visibility, at least in this form, therefore I am not in favor of publishing the work.

Author Response

Thanks for your feedback.

We have

  1. restructured the article,
  2. abandoned the third sub-level in the introduction,
  3. shortened the introduction altogether,
  4. as well as shortened the methods section,
  5. and have expanded and extended the results section and the discussion.

We hope that we have met your recommendations in this way.

Round 3

Reviewer 5 Report

Dear authors, you did a good job modifying the paper. Only one aspect consider that must be solved: The sources from appendix are too many.  Either You cite them in refence part, either you give up on them.

Author Response

Dear Reviewer 5,

thanks for your feedback. Your are right, we realised that our appendixes are too long. We have deleted them and added the data availability statement at the end of the manuscript.

Best regards